# Construction and Functional Analysis of the ceRNA Regulatory Network Associated with Muscle Development in Shaanbei White Cashmere Goats

**DOI:** 10.3390/ani15243568

**Published:** 2025-12-11

**Authors:** Lina Liu, Fenghong Wang, Long Zhou, Zhaofei Ren, Shutao Shang, Lei Qu, Haijing Zhu, Lei Zhang

**Affiliations:** 1Shaanxi Provincial Engineering and Technology Research Center of Cashmere Goats, College of Advanced Agricultural Sciences, Yulin University, Yulin 719000, China; lina0823724@163.com (L.L.); 19561304025@163.com (L.Z.); 15891221869@163.com (Z.R.); 15891171006@163.com (S.S.); ylqulei@126.com (L.Q.); haijingzhu@yulinu.edu.cn (H.Z.); 2College of Modern Agriculture, Yulin University, Yulin 719000, China; nmgcfwfh@163.com

**Keywords:** muscle generation, MSTRG.5182.1, cashmere goat, LncRNAs

## Abstract

This study used RNA sequencing to investigate the role of long non-coding RNAs (lncRNAs) in muscle growth of cashmere goats. Muscle growth is vital for meat quality and yield, but its molecular regulation is not fully understood. We analyzed longissimus dorsi muscles from cashmere goats at four developmental stages (120 days fetal, 1 month, 3 months, and 10 months old) and identified 3480 lncRNAs, including 1141 newly discovered. This study found that a novel lncRNA (MSTRG.5182.1) plays a key role in muscle development by regulating the chi-miR-424-5p/*IKBKG* pathway. This discovery provides insights into muscle development in cashmere goats and suggests potential molecular targets for enhancing mutton quality and meat production, promoting molecular breeding and meat quality improvement in cashmere goats.

## 1. Introduction

In recent years, China’s economy has continued to grow rapidly, and its per capita disposable income has been steadily increasing. Consequently, residents’ consumption of high-quality meat has been increasing. Among various types of meat, goat meat is highly favored for its tender texture, unique flavor, and high protein content. However, the level of muscle development directly determines the meat yield and quality, which has become a key bottleneck restricting the development of the goat meat industry. Livestock and poultry muscle growth is regulated by a combination of factors, including the genotype, nutritional supply, and environmental influences, with complex interaction networks existing among these factors [1,2,3]. Although the *Shaanbei White Cashmere goat* has breed advantages, such as a tolerance to coarse feed and strong resilience, current populations commonly face issues like a small muscle fiber diameter, insufficient tenderness, and abnormal fat deposition. The fundamental reason is that the regulatory mechanisms of genes related to muscle growth and development are not yet clear, and traditional breeding methods cannot achieve breakthrough improvements in meat performance. Therefore, improving meat performance has become an urgent issue to be addressed in animal husbandry.

Non-coding RNAs (such as miRNAs, lncRNAs, and circRNAs) can precisely regulate key biological processes in myocytes, including proliferation and differentiation, muscle fiber type conversion, and lipid metabolism, at the post-transcriptional and epigenetic levels by targeting mRNAs or chromatin remodeling complexes [4,5]. In particular, the discovery of competitive endogenous RNA (ceRNA) networks has revealed a new mechanism by which RNA molecules cross-regulate each other through shared miRNA response elements [6]. This has provided a novel perspective for deciphering the molecular regulatory networks of muscle development.

Muscle growth and development are key factors determining the meat production performance of livestock and poultry [7]. Previous studies have found 285 differentially expressed genes when comparing high- and low-intramuscular-fat (IMF) groups in the longissimus dorsi muscle of *Ningxiang pigs* [8]. Similarly, whole-transcriptome analyses in cattle and chickens have also identified a large number of non-coding RNAs. However, studies on their mechanisms and modifications in muscle development remain relatively limited [9,10]. An analysis of the longissimus dorsi, subcutaneous fat, and perirenal fat tissues in *Qinchuan beef cattle* revealed 105 and 76 miRNAs differentially expressed between the longissimus dorsi and subcutaneous fat and perirenal fat, respectively, with 40 upregulated and 14 downregulated in the muscle [11]. Compared with other species, whole-transcriptome analyses in sheep still require further in-depth studies of RNA editing and modifications. Future research should focus more on these aspects to further elucidate the role of RNA editing and modifications in sheep muscle development.

The *Shaanbei White Cashmere goat* is mainly distributed in the northern region of Shaanxi Province. Through years of breeding, it has developed into a breed primarily utilized for wool as well as meat production [12]. In this study, 12 longissimus dorsi muscle tissue samples from *Shaanbei White Cashmere goats* were used as biological samples, taken at a fetal age of 120 days and at 1 month, 3 months, and 10 months of age [13,14,15], covering 14 muscle tissue development stages. An RNA-seq analysis detected a total of 3480 differentially expressed lncRNAs (DELs), and a bioinformatics analysis identified a series of candidate lncRNAs. The lncRNA associated with cashmere goat muscle growth and development (MSTRG.5182.1) was identified, and its biological function in myogenesis was explored. This study aims to reveal the roles and mechanisms of lncRNAs in cashmere goat myogenesis, laying a foundation for improving the mutton quality, targeted selection, and molecular breeding of meat goats.

## 2. Materials and Methods

### 2.1. Sample Preparation

Longissimus dorsi muscle (LDM) tissues were collected from healthy female *Shaanbei White Cashmere goats* at different developmental stages. Specifically, samples were obtained from goats at fetal age (120 days post-gestation, 120 dPG), 1 month old (1 mo), 3 months old (3 mo), and 10 months old (10 mo). The goats were raised at the Yulin University experimental sheep farm (Yulin, Shaanxi, China) under the care of the same dedicated staff to ensure consistent feeding and management conditions. Using sterile surgical instruments, LDM tissues were rapidly excised and immediately frozen in liquid nitrogen. The samples were then stored at −80 °C for subsequent RNA extraction. Each developmental stage included three biological replicates. All cashmere goats were humanely anesthetized prior to tissue collection to minimize suffering. Goats in each group were of similar weight and body size to ensure comparability across the different developmental stages.

### 2.2. RNA Extraction, Library Preparation, and Sequencing

RNA sequencing (RNA-seq) analysis was performed on a total of 12 LDM tissue samples, with three biological replicates collected at four distinct muscle development stages: 120-day fetal age, 1 month, 3 months, and 10 months. Total RNA was extracted from the samples following the manufacturer’s protocol. To enrich the coding RNA and non-coding RNA (ncRNA), ribosomal RNA was removed from the total RNA. The RNA integrity was assessed using the RNA Integrity Number (RIN), and only samples with an RIN ≥ 8.0 were included in the analysis to ensure high-quality RNA samples. The purified RNA was then randomly fragmented into shorter pieces and used as a template to synthesize the first strand of cDNA using random hexamer primers. Subsequently, the second strand of cDNA was synthesized by adding a buffer, dNTPs (with dUTP replacing dTTP), RNase H, and DNA polymerase I. The cDNA was purified using the QiaQuick PCR kit (Qiagen, Hilden, Germany) and eluted with an EB buffer. After end repair, A-tailing, and adapter ligation, the second strand was degraded using UNG enzyme. The cDNA fragments were size-selected by agarose gel electrophoresis and PCR amplified. The prepared sequencing library was then sequenced using the Illumina HiSeqTM 4000 (Illumina, California, CA, USA) platform.

The raw sequencing reads were subjected to quality control using fastp [16] to remove adapter-containing reads, reads with more than 10% N content, reads consisting entirely of A bases, and low-quality reads (where bases with Q ≤ 20 accounted for over 50% of the read), resulting in clean reads. HISAT2 [17] was used for reference genome-based alignment, and transcripts were reconstructed using StringTie, retaining those with a length of ≥200 bp and at least two exons. The coding potential of novel lncRNA transcripts was predicted using CPC (version 0.9-r2) [18], CNCI (version 2) [19], and FEELnc (version v0.2) [20] software. The intersection of the results from these three tools was taken as the final lncRNA prediction.

### 2.3. Identification of LncRNA and Screening of DEls

The classification of novel lncRNAs was conducted based on their genomic positions relative to protein-coding genes. The expression levels of lncRNAs in the samples were analyzed by calculating the sequencing depth and transcript length to obtain the fragments per kilobase of transcript per million mapped reads (FPKM) values of lncRNAs. The expression distribution of different samples was displayed using expression distribution plots. Principal component analysis (PCA) was performed using R (http://www.r-project.org/, accessed on 17 June 2024) to study the distance relationships between samples. The Pearson correlation coefficient between each pair of samples was calculated and shown in a heatmap to depict correlations. Differential expression analysis was performed on the read count data obtained from lncRNA expression analysis. Read counts were normalized using the DESeq2 (version 1.20.0) [21] software. *p*-values were calculated for hypothesis testing, and multiple testing correction was applied to obtain false discovery rate (FDR) values (<0.05).

### 2.4. Prediction and Functional Analysis of miRNAs Targeted by DEls and Target Genes of Key miRNAs

To elucidate the potential functions of DEls in muscle development, we conducted Gene Ontology (GO) and Kyoto Encyclopedia of Genes and Genomes (KEGG) enrichment analyses on their target genes using OmicShare Tools on the Kiddio Bioinformatics Cloud Platform (https://www.omicshare.com/) (accessed on 24 July 2024). These analyses were performed with a significance threshold of *p* ≤ 0.05 for enrichment. We used TargetScan [22,23,24] and miRanda [25,26] to predict miRNAs targeted by lncRNAs. TargetScan predicts miRNA targets based on the seed region of the miRNA, with a higher score indicating a greater likelihood of interaction (threshold set at 50). miRanda relies on the binding free energy between lncRNA and miRNA, with more negative values indicating stronger binding and functionality (threshold set at −10). By combining the predictions from these two tools, we identified lncRNA–miRNA targeting relationships. Similarly, we predicted target genes of key miRNAs to obtain miRNA–mRNA targeting relationships. These predictions were used to construct a competitive endogenous RNA (ceRNA) regulatory network related to muscle growth and development. This network provides insights into the potential roles of lncRNAs and miRNAs in regulating muscle development processes.

### 2.5. Validation of Transcriptome Sequencing Data by qRT-PCR

To validate the accuracy of the transcriptome sequencing data, we randomly selected four DEl molecules and their target genes for verification using real-time quantitative PCR (qRT-PCR). Total RNA samples from the original samples were reverse transcribed into cDNA using a reverse transcription kit (Tiangen Biochemical Technology Co., Ltd., Beijing, China) according to the manufacturer’s instructions [27]. β-actin was used as the internal reference gene [28]. Primers were synthesized by Shanghai Shenggong Bioengineering Co., Ltd. (Shanghai, China), and the sequence information is provided in Table 1. The qRT-PCR reaction system included 10 µL of SYBR^®^ Premix En TaqTM II (2×), 0.8 µL of forward primer, 0.8 µL of reverse primer, 0.4 µL of ROX Reference Dye (50×), 2 µL of cDNA template, and 0.6 µL of ddH_2_O.

Similarly, four differentially expressed miRNAs were randomly selected for validation using qRT-PCR. Primers were designed according to the stem–loop primer method [29], and total RNA from the original samples was reverse transcribed into cDNA using the same reverse transcription kit (Tiangen Biochemical Technology Co., Ltd., Beijing, China), following the manufacturer’s instructions. U6 was used as the internal reference, and primers were synthesized by Sangon Biotech Co., Ltd., Shanghai, China. The sequence information is provided in Table 2. The qRT-PCR reaction system consisted of 10 µL of SYBR^®^ Premix En TaqTM II (2×), 0.8 µL of forward primer, 0.8 µL of reverse primer, 0.4 µL of ROX Reference Dye (50×), 2 µL of cDNA template, and 0.6 µL of ddH_2_O. Each target gene and reference gene were run in triplicate for each sample. The qRT-PCR program was as follows: pre-denaturation at 95 °C for 30 s, followed by 40 cycles of 95 °C for 5 s and 57.5 °C for 30 s. The melting curve program was 94 °C for 1 min 30 s and 60 °C for 3 min. The results were analyzed using the 2^−ΔΔCt^ method [30] and plotted with GraphPad Prism 8 software.

### 2.6. Luciferase Assays to Validate miRNA–lncRNA Interactions

To elucidate the functional role of chi-miR-424-5p in muscle development, we synthesized chi-miR-424-5p mimics using a kit from Shanghai Hanbio Biotechnology Co., Ltd., Shanghai, China. We generated the pSI-Check2-MSTRG.5182.1-WT and pSI-Check2-IKBKG-WT constructs by inserting fragments of MSTRG.5182.1 and *IKBKG* containing miRNA binding sites into the 3′ end of the Renilla luciferase gene in the pSI-Check2 vector (Promega) (Hanbio Biotechnology Co. Ltd., Shanghai, China). To produce mutant constructs, we mutated the miRNA binding sequences to complementary sequences, resulting in the pSI-Check2-MSTRG.5182.1-MUT and pSI-Check2-IKBKG-MUT constructs.

For luciferase assays, HEK 293T cells were transfected with miRNA mimics along with the pSI-Check2-MSTRG.5182.1-WT (or pSI-Check2-IKBKG-WT) or mutant pSI-Check2-MSTRG.5182.1-MUT (or pSI-Check2-IKBKG-MUT) reporter plasmids. Forty-eight hours after transfection, luciferase activity was measured using the Dual-Luciferase Reporter Assay System (Promega). The relative luciferase activity was calculated by comparing the ratio of firefly to Renilla luciferase activities.

Cell Transfection Procedure: The 293T cells were prepared for transfection by culturing to 50–70% confluence in a 96-well plate. The transfection mixture was prepared by combining 10 μL of DMEM containing 0.16 μg of the g-MSTRG.5182.1 plasmid and 5 pmol of chi-miR-424-5p or negative control with 10 μL of DMEM containing 0.3 μL of transfection reagent (used to introduce constructs into cells) (from Hanbio Biotechnology Co. Ltd., Shanghai, China, concentration 0.8 mg/mL). After incubating at room temperature for 20 min, the mixture was added to the cells and incubated at 37 °C with 5% CO_2_ for 6 h. Cells were then collected 48 h post-transfection for analysis.

This procedure was used to validate the interaction between chi-miR-424-5p and its target MSTRG.5182.1 and *IKBKG*, providing insights into the regulatory mechanisms underlying muscle development.

### 2.7. Statistical Analysis

Data are expressed as mean ± standard deviation (SD). Statistical analyses and figure construction were conducted using GraphPad Prism 8.0.1 (GraphPad 8.0.1 Software, San Diego, CA, USA) [31]. One-way analysis of variance (ANOVA) was employed to determine the significance of differences among groups. The significance level is set as *** *p* < 0.001 indicates a highly significant difference, ** *p* < 0.01 indicates a moderately significant difference, * *p* < 0.05 indicates a significant difference [14]. Additionally, all transcriptome data analyses were performed on the Omic Smart platform (https://www.omicsmart.com/#/, accessed on 17 June 2024), which provided a comprehensive suite of tools for data analysis.

## 3. Results

### 3.1. Sequencing Data: Quality Assessment

A total of 1,072,767,186 raw sequencing reads were generated from the libraries constructed for the 12 samples. After stringent filtering to remove low-quality reads and adapter sequences, 1,068,657,830 high-quality reads were retained. The quality metrics for each sample were as follows: Q20 ≥ 97%, Q30 ≥ 93%, an error rate ≤ 0.09%, and a stable GC content. More than 76% of the high-quality reads were successfully aligned to the goat reference genome (Table 3). These results indicate that the sequencing data were of high quality and suitable for further analysis.

### 3.2. Characteristics of lncRNAs Involved in Muscle Formation

Based on the alignment results of all reads mapped to the genome (Total_Mapped reads), the distribution of reads across the reference genome was analyzed. The clean reads from the 12 samples primarily aligned to known exons, with relatively low and stable proportions in introns and intergenic regions. These results indicate the effective removal of rRNA and complete genome annotation, confirming that the data quality is suitable for the lncRNA analysis. A few samples exhibited slightly higher intron proportions, which may suggest the presence of novel transcripts or alternative splicing events. These can be further explored through differential expression analysis and novel lncRNA identification (Figure 1A).

Using StringTie for transcript reconstruction, the coding potential of novel transcripts was assessed with three software programs: CPC (version 0.9-r2), CNCI (version 2), and FEELnc (version v0.2). Transcripts that lacked coding potential were considered reliably predicted lncRNAs (Figure 1B). Based on their genomic locations relative to protein-coding genes, the novel lncRNAs were classified into five major categories: intergenic (52.4%), antisense (13.4%), sense-overlapping (7.2%), intronic (11.6%), and bidirectional (10.22%) (Figure 1C).

The distribution of gene expression levels across all samples showed high consistency (Figure 1D). Violin plots were used to depict the expression level distribution of transcripts in each sample, highlighting differences in transcript expression among samples (Figure 1E). The principal component analysis (PCA) was performed using R (http://www.r-project.org/) (accessed on 17 June 2024), revealing significant differences in expression patterns among different sample groups (Figure 1F). The Pearson correlation coefficient was calculated for the expression levels of each lncRNA (the entire lncRNA set) between any two samples. The expression patterns among samples were highly similar, with correlation coefficients generally close to one (Figure 1G).

### 3.3. Differential Expression Analysis of lncRNAs During Myogenesis

In this study, a comprehensive analysis of lncRNA expression during myogenesis was conducted. A total of 643 lncRNAs were identified across the comparison groups (Figure 2A). The differential expression analysis revealed distinct patterns of lncRNA regulation at various stages of muscle development. Specifically, in the CG-A-1 vs. CG-B-3 group, we identified 1 upregulated and 11 downregulated lncRNAs; in the CG-A-1 vs. CG-C-10 group, 3 upregulated and 15 downregulated lncRNAs were detected; in the CG-A-1 vs. CG-D-120 group, 48 upregulated and 121 downregulated lncRNAs were found; in the CG-B-3 vs. CG-C-10 group, 6 upregulated and 22 downregulated lncRNAs were observed; in the CG-B-3 vs. CG-D-120 group, 76 upregulated and 91 downregulated lncRNAs were noted; and in the CG-C-10 vs. CG-D-120 group, 115 upregulated and 134 downregulated lncRNAs were identified (Figure 2B). To identify DEls, we employed DESeq2 with stringent criteria of an FDR < 0.05 and |log2FC| > 1. This analysis resulted in the identification of 272 DElncRNAs across the four comparison groups (Figure 2C). These DEls likely play crucial roles in the regulation of muscle development. Further, we identified a total of 3339 target genes that were significantly associated with the DEls. To explore the potential functions of these target genes, a GO enrichment analysis was performed. The top 30 GO terms were selected based on the *p*-value (Figure 2D). Notably, the analysis revealed several muscle development-related terms, including muscle structure development (GO:0061061), contractile fiber (GO:0043292), and cell differentiation (GO:0030154). Additionally, the target genes of numerous DEls were enriched in 316 KEGG pathways. The top 30 KEGG pathways were selected according to *p*-values. Muscle development-related pathways included the MAPK signaling pathway, extracellular matrix–receptor interaction, adrenergic signaling in cardiomyocytes, and the calcium signaling pathway, among others (Figure 2E). These findings provide valuable insights into the molecular mechanisms underlying muscle development and highlight the potential regulatory roles of lncRNAs in this process.

### 3.4. Identification of miRNAs Targeted by lncRNAs in Muscle Growth and Development

To identify miRNAs targeted by lncRNAs related to muscle growth and development, we performed a differential analysis on miRNAs using the edgeR software (version 3.12.1), with the dispersion set to 0.01 and other parameters at their default values. This analysis revealed the following differentially expressed miRNAs across the comparison groups: In the CG-A-1 vs. CG-B-3 group, 42 upregulated and 129 downregulated miRNAs were identified. In the CG-A-1 vs. CG-C-10 group, 107 upregulated and 163 downregulated miRNAs were identified. In the CG-A-1 vs. CG-D-120 group, 259 upregulated and 183 downregulated miRNAs were identified. In the CG-B-3 vs. CG-C-10 group, 87 upregulated and 93 downregulated miRNAs were identified. In the CG-B-3 vs. CG-D-120 group, 314 upregulated and 166 downregulated miRNAs were identified. In the CG-C-10 vs. CG-D-120 group, 307 upregulated and 234 downregulated miRNAs were identified (Figure 3A). By selecting significantly upregulated and downregulated miRNAs from these comparison groups, we identified a total of 27 significantly different miRNAs across the four comparison groups (Figure 3B). Figure 3 focuses on the interactions between miRNAs and ceRNAs, which play a pivotal role in post-transcriptional regulation. Although lncRNAs can modulate mRNA levels, the current analysis was restricted to the miRNA–ceRNA axis in order to provide a focused examination of this specific regulatory mechanism. Future studies will explore the broader implications of lncRNAs for 1 mRNA expression levels.

### 3.5. Analysis of miRNA and ceRNA Regulatory Networks

To construct the ceRNA regulatory network, we calculated the Spearman rank correlation coefficients between the miRNAs and candidate ceRNAs for the identified targets, screening for target gene pairs with a correlation coefficient less than or equal to −0.7. The expression levels of ceRNAs competing for the same miRNA are positively correlated. We then calculated the Pearson correlation coefficients for the expression levels between the ceRNA pairs obtained in the previous step, selecting ceRNA pairs with a correlation coefficient of 0.9 or higher as potential ceRNA pairs. Based on these results, we further used the hypergeometric cumulative distribution function test [32] to screen for ceRNA pairs with a *p*-value less than 0.05 as the final ceRNA pairs (Figure 3C). In the ceRNA regulatory network, the connectivity of an RNA molecule is defined as the number of miRNAs that have targeting relationships with it. The higher the connectivity of an RNA molecule, the stronger its potential regulatory capacity. Figure 3D illustrates the mRNA connectivity, while Figure 3E displays the lncRNA connectivity. For the top three lncRNAs in regard to connectivity and the top six mRNAs in regard to connectivity, we plotted a Sankey diagram illustrating their miRNA-targeting regulatory relationships (Figure 3F) and further constructed their ceRNA network (Figure 4).

### 3.6. Validation of ceRNA Regulatory Network by qRT-PCR

To validate the ceRNA regulatory network identified in our study, we selected the top three significantly different RNA molecules from the ceRNA network. Figure 5A displays the absolute expression levels of these significantly different RNA molecules. The expression levels were further verified using qRT-PCRs. The results shown in Figure 5B indicate that the qRT-PCR trends of the RNA molecules are consistent with their absolute quantification, thereby confirming the reliability of the sequencing results.

### 3.7. The Characterization of the MSTRG.5182.1-chi-miR-424-5p/IKBKG Regulatory Axis in Muscle Development

To elucidate the regulatory mechanisms underlying muscle growth and development, we conducted a comprehensive analysis of differentially expressed genes by mapping them to the GO database and calculating the number of differentially expressed genes for each GO term. This analysis generated a statistical overview of genes with specific GO functions. We then employed the hypergeometric test to identify GO terms that were significantly enriched in the differentially expressed genes compared to the background. In the lncRNA–mRNA GO analysis, the differentially expressed genes were significantly enriched in processes such as cell growth, proliferation, differentiation, and tissue development (Figure 6A). A pathway enrichment analysis was performed based on KEGG pathways, using the hypergeometric test to identify pathways that were significantly enriched in the differentially expressed genes compared to the overall background. In the lncRNA–mRNA pathway analysis, the differentially expressed genes were significantly enriched in the MAPK signaling pathway and the calcium signaling pathway (Figure 6B). By integrating functional and significance predictions, we identified the lncRNA–mRNA pair MSTRG.5182.1–*IKBKG* as a key regulatory pair. This analysis culminated in the ceRNA network diagram depicted in Figure 6C. Combining this with the connectivity analysis from the experimental results in Figure 3F, we predicted that MSTRG.5182.1 participates in the regulation of muscle growth and development by targeting the chi-miR-424-5p/*IKBKG* signaling axis.

### 3.8. Functional Analysis of the MSTRG.5182.1-chi-miR-424-5p/IKBKG Regulatory Axis

We conducted a preliminary validation of the two targeting axes, MSTRG.5182.1–chi-miR-424-5p and chi-miR-424-5p–*IKBKG*. qRT-PCR results revealed that the two targeting axes exhibited opposite trends, consistent with the ceRNA network regulatory relationship (Figure 7A,B). To predict the existence of binding sites between chi-miR-424-5p and both MSTRG.5182.1 and *IKBKG*, we utilized TargetScan and miRanda. Subsequently, mutant vectors were constructed to verify specific binding sites (Figure 7C,D).

The results demonstrated that, compared with the negative control (NC) group, chi-miR-424-5p significantly reduced the luciferase expression in muscle tissue for MSTRG.5182.1 (*p* < 0.001), indicating the presence of a binding site between chi-miR-424-5p and MSTRG.5182.1. Following the mu1 mutation, chi-miR-424-5p failed to downregulate the luciferase expression in MSTRG.5182.1-mut1 (*p* > 0.05), confirming a successful mutation and verifying that chi-miR-424-5p can bind to MSTRG.5182.1.

Similarly, compared with the NC group, chi-miR-424-5p significantly reduced the luciferase expression in *IKBKG* cells (*p* < 0.001), indicating a binding site between chi-miR-424-5p and *IKBKG*. After the mu1 mutation, chi-miR-424-5p failed to downregulate the luciferase expression in *IKBKG*-mut1 (*p* > 0.05), confirming a successful mutation and verifying that chi-miR-424-5p can bind to *IKBKG* (Figure 7E,F). These findings further demonstrate that the MSTRG.5182.1–chi-miR-424-5p/*IKBKG* axis has a targeting relationship.

We speculate that MSTRG.5182.1 may affect *IKBKG* expression through a ceRNA mechanism by competitively binding with chi-miR-424-5p (Figure 8), thereby regulating muscle growth and development in cashmere goats.

## 4. Discussion

This study employed high-throughput RNA sequencing technology to identify 3480 DEls in the longissimus dorsi muscle of *Shaanbei White Cashmere goats* across four key developmental stages: a fetal age of 120 days, 1 month, 3 months, and 10 months. This comprehensive dataset significantly enhances our understanding of the non-coding RNA regulatory network during goat muscle development. Compared to the 548 lncRNAs recently identified in *Chuanzhong black goats* [33], our study provides a more complete developmental time series, offering a more comprehensive perspective on the dynamic regulation of lncRNAs in muscle development.

Through a classification analysis of the newly identified lncRNAs, we found that intergenic lncRNAs account for 52.4% and antisense lncRNAs account for 13.4%, which aligns with findings from other livestock studies [34]. Notably, MSTRG.5182.1, a newly discovered lncRNA, exhibited significant expression differences during muscle development, suggesting a key regulatory role in myogenesis.

The lncRNA–miRNA–mRNA ceRNA network constructed in this study reveals the molecular mechanism by which MSTRG.5182.1 regulates *IKBKG* expression through competitively binding chi-miR-424-5p. This finding is highly consistent with ceRNA regulatory patterns observed in the muscle development of other livestock. For example, in Jianchang black goats, Han et al. [35] found that lncRNA XR_001296113.2 regulates *PDLIM7* expression by sponging chi-miR-1296, while XR_001917947.1 affects smad3 expression by competitively binding chi-miR-30b-3p. *PDLIM7*, a member of the PDZ-LIM protein family, is important for skeletal muscle development, and the role of smad3 in skeletal muscle regeneration is well established.

Our research further expands the ceRNA regulatory network of goat muscle development, particularly revealing a novel function of chi-miR-424-5p. Dual-luciferase reporter assays confirmed the direct interaction between MSTRG.5182.1 and chi-miR-424-5p, as well as the targeted regulatory relationship of chi-miR-424-5p with IKBKG. This discovery provides new information for understanding the interactions among non-coding RNAs and opens new avenues for exploring the complex regulatory mechanisms of muscle development. Chi-miR-424-5p not only functions through the traditional mRNA degradation mechanism but also acts as a central node in the ceRNA network to regulate muscle development. Recent studies have shown that miR-424-5p reduces protein synthesis during skeletal muscle atrophy by inhibiting rRNA synthesis, and its rodent homolog miR-322-5p is downregulated in muscle hypertrophy models [36]. Connolly et al. [37] found that miR-424-5p inhibits the protein synthesis capacity by targeting key rRNA transcription factors such as PolR1A and UBTF, leading to muscle atrophy.

However, our study reveals, for the first time, the ceRNA regulatory mechanism of chi-miR-424-5p in the muscle development of Shaanbei White Cashmere goats, suggesting that its function may be species-specific and dependent on developmental stages. During muscle development, chi-miR-424-5p may participate in fine-tuning muscle growth through different regulatory patterns. This dual regulatory mode reflects the complex roles of miRNAs in muscle development, providing new insights for a deeper understanding of the molecular regulation of muscle growth. Our findings also indicate that the functions of miRNAs may be far more complex than previously recognized and require in-depth study within the context of specific biological backgrounds and species characteristics.

This study identified *IKBKG* as a direct target gene of chi-miR-424-5p, revealing the important role of the NF-κB signaling pathway in goat muscle development. The NF-κB pathway plays a dual role in muscle development: the classical pathway maintains the muscle stem cell pool by promoting myoblast proliferation and inhibiting premature differentiation, while the non-classical pathway is activated in the later stages of differentiation to promote mitochondrial biogenesis and myotube maturation [38]. Research by Bakkar and colleagues indicated that conventional NF-κB signaling is downregulated during differentiation, whereas the alternative members IKKα, RelB, and p52 are induced in the later stages of myogenesis [39]. Our study found that MSTRG.5182.1 may promote the activation of the NF-κB pathway by relieving the inhibition of IKBKG by chi-miR-424-5p, thereby regulating muscle development. IKBKG (also known as NEMO) is a key regulatory subunit of the IKK complex and is crucial for the activation of the classical NF-κB pathway [40]. During muscle development, the precise regulation of the NF-κB signaling pathway is important for maintaining the balance between muscle stem cell proliferation and differentiation [41,42].

Notably, this study systematically revealed the lncRNA regulatory network during muscle development in the *Shaanbei White Cashmere goat*, particularly uncovering the novel MSTRG.5182.1/chi-miR-424-5p/*IKBKG* ceRNA regulatory axis. Compared with existing studies, our research has the following characteristics: (1) it covered a more complete developmental time series, from the embryonic stage to 10 months of age; (2) it experimentally confirmed the molecular mechanism of the ceRNA network; and (3) it revealed a new regulatory pattern of the NF-κB pathway in the muscle development of *Shaanbei White Cashmere goats*. While our findings provide an important theoretical basis for understanding the molecular mechanisms of muscle development in cashmere goats, they are limited by the lack of in vivo functional validation.

These findings not only provide an important theoretical basis for understanding the molecular mechanisms of muscle development in cashmere goats but also offer potential molecular markers for molecular breeding and meat quality improvement. MSTRG.5182.1, as a newly discovered muscle development-related lncRNA, may have an expression pattern associated with key traits, such as the muscle growth rate and meat quality, and could serve as a candidate marker gene for molecular breeding in cashmere goats. Future research could further explore the conservation of this regulatory network across different goat breeds and its potential applications in muscle-related diseases. Additionally, the technical approaches and analytical methods established in this study could serve as references and guidance for studying meat quality traits in other livestock.

## 5. Conclusions

In this study, we employed RNA-seq technology in conjunction with molecular biology experiments to systematically elucidate the lncRNA regulatory network involved in the muscle development of Shaanbei White Cashmere goats. Our research particularly focused on the molecular mechanism underlying the MSTRG.5182.1/chi-miR-424-5p/*IKBKG* ceRNA regulatory axis. The findings of this study offer novel insights into the molecular regulatory mechanisms governing muscle development in livestock. Moreover, they provide a robust theoretical framework for advancing molecular breeding strategies in cashmere goats and improving meat quality traits. By expanding the ceRNA regulatory network associated with goat muscle development, our research not only contributes to a deeper understanding of muscle growth regulation but also serves as a valuable reference for similar studies in other livestock species.

## Figures and Tables

**Figure 1 animals-15-03568-f001:**
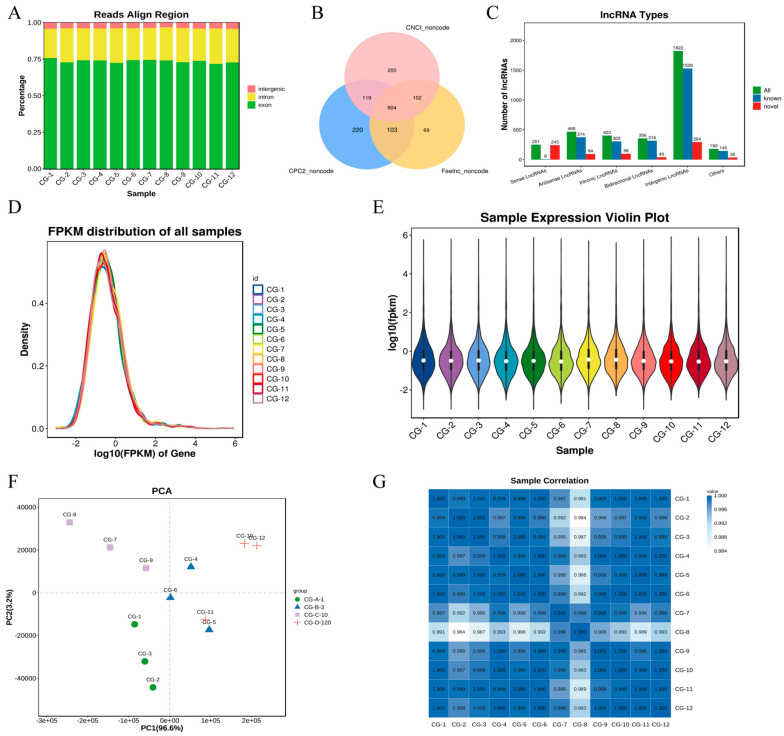
Characterization of lncRNAs involved in muscle formation. CG-A-1 represents 1-month-old female *Shaanbei White Cashmere goats*, CG-B-3 represents 3-month-old female *Shaanbei White Cashmere goats*, CG-C-10 represents 10-month-old female *Shaanbei White Cashmere goats*, and CG-D-120 represents a female *Shaanbei White Cashmere Goat kid* at 120 days of embryonic age. (**A**) Comparison of reference area; (**B**) classification diagram of Coding Potential Calculator 2 (CPC2), Coding Non-Coding Index (CNCI), and Function and Expression Long non-coding (FEELnc); (**C**) lncRNA type statistics; (**D**) distribution of lncRNA expression abundance; (**E**) violin plot of lncRNA expression levels; (**F**) principal component analysis (PCA) of lncRNA samples; and (**G**) correlation heatmap of lncRNA samples.

**Figure 2 animals-15-03568-f002:**
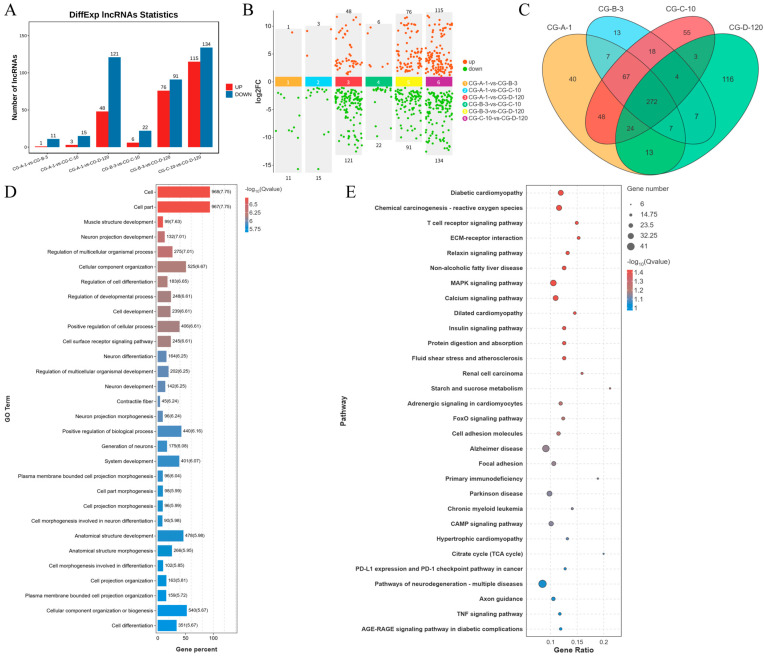
Comprehensive analysis of DELs during myogenesis: CG-A-1 represents 1-month-old female *Shaanbei White Cashmere goats*, CG-B-3 represents 3-month-old female *Shaanbei White Cashmere goats*, CG-C-10 represents 10-month-old female *Shaanbei White Cashmere goats*, and CG-D-120 represents a female *Shaanbei White Cashmere Goat kid* at 120 days of embryonic age. (**A**) Differential lncRNA statistical chart; (**B**) LncRNA multi-group differential scatter plot; (**C**) DEL Venn diagram; (**D**) DEL target gene GO enrichment analysis; and (**E**) DEL target gene KEGG enrichment analysis.

**Figure 3 animals-15-03568-f003:**
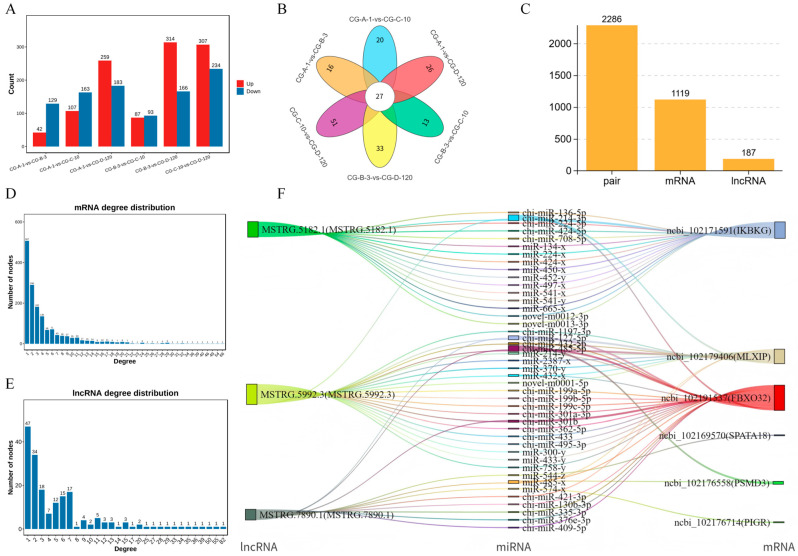
In-depth analysis of miRNA and ceRNA regulatory networks: CG-A-1 represents 1-month-old female *Shaanbei White Cashmere goats*, CG-B-3 represents 3-month-old female *Shaanbei White Cashmere goats*, CG-C-10 represents 10-month-old female *Shaanbei White Cashmere goats*, and CG-D-120 represents a female *Shaanbei White Cashmere Goat kid* at 120 days of embryonic age. (**A**) Statistical chart of miRNA differences for each comparison group; (**B**) differential miRNA analysis between each comparison group; (**C**) LncRNA-mRNA statistical chart; (**D**) mRNA connectivity distribution map; (**E**) LncRNA connectivity distribution map; and (**F**) regulatory relationship Sankey diagram.

**Figure 4 animals-15-03568-f004:**
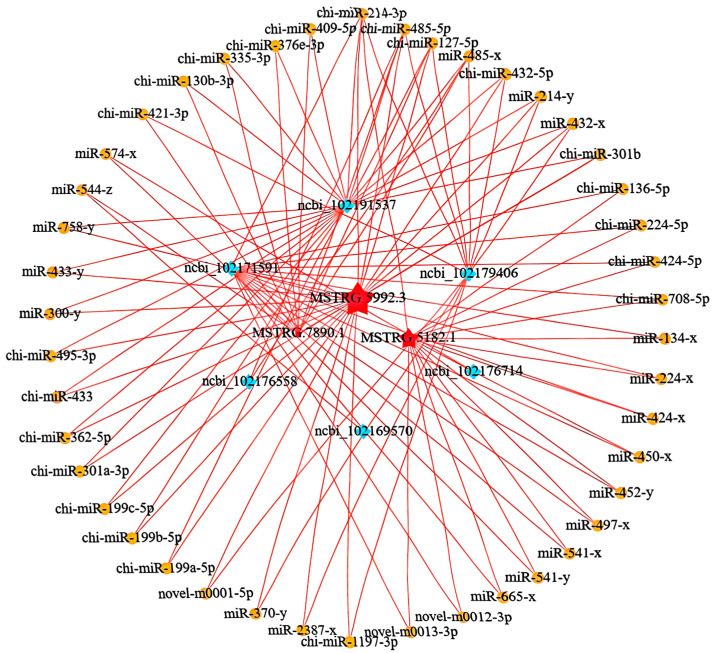
Visualization of the ceRNA regulatory network.

**Figure 5 animals-15-03568-f005:**
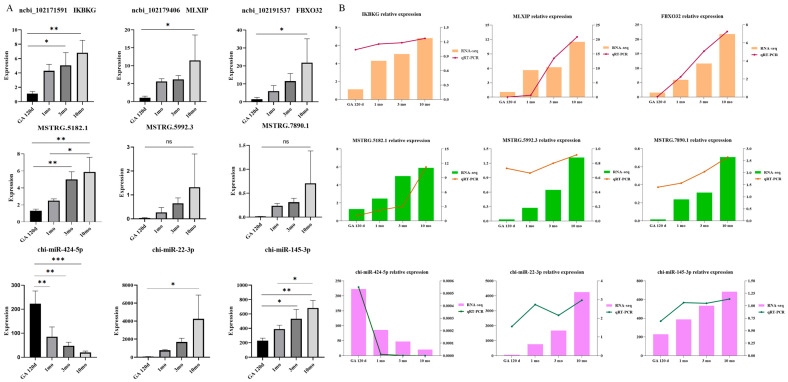
Validation of experimental results: The significance level is set as *** *p* < 0.001 indicates a highly significant difference, ** *p* < 0.01 indicates a moderately significant difference, * *p* < 0.05 indicates a significant difference (**A**) absolute quantitative significance analysis and (**B**) qRT-PCR experiment verify the accuracy of sequencing data.

**Figure 6 animals-15-03568-f006:**
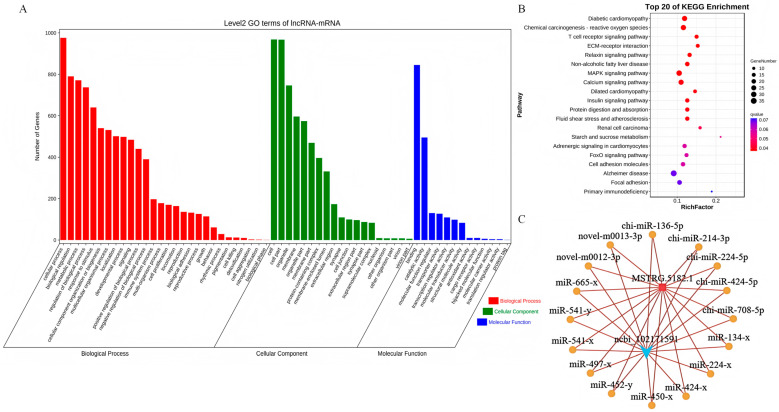
Functional enrichment analysis and ceRNA network construction: (**A**) LncRNA–mRNA GO enrichment analysis; (**B**) LncRNA–mRNA pathway enrichment analysis; and (**C**) final ceRNA network diagram.

**Figure 7 animals-15-03568-f007:**
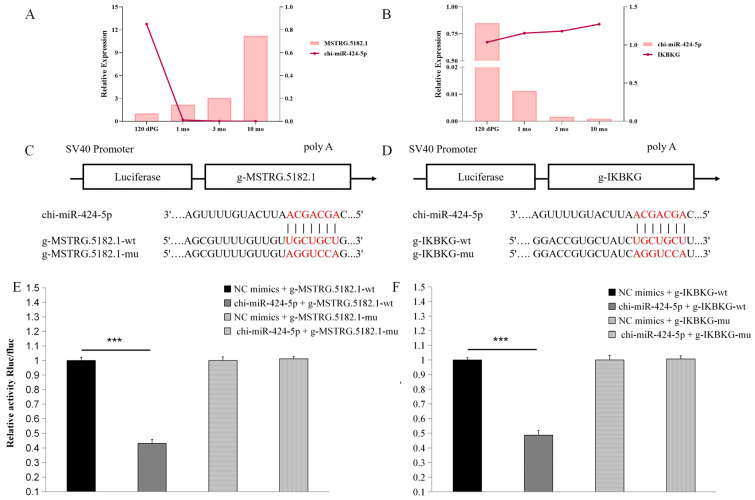
Preliminary validation of the MSTRG.5182.1-chi-miR-424-5p/*IKBKG* regulatory axis function: The significance level is set as *** *p* < 0.001 indicates a highly significant difference (**A**) qRT-PCR validation of lncRNA-miRNA regulatory relationships; (**B**) qRT-PCR verification of miRNA–mRNA regulatory relationships; (**C**) schematic diagram of the binding site between chi-miR-424-5p and g-MSTRG.5182.1; (**D**) schematic diagram of chi-miR-424-5p binding to g-IKBKG target site; (**E**) dual-luciferase reporter assay detects the interaction between chi-miR-424-5p and g-MSTRG.5182.1; and (**F**) dual-luciferase reporter gene assay detects the interaction between chi-miR-424-5p and g-IKBKG.

**Figure 8 animals-15-03568-f008:**
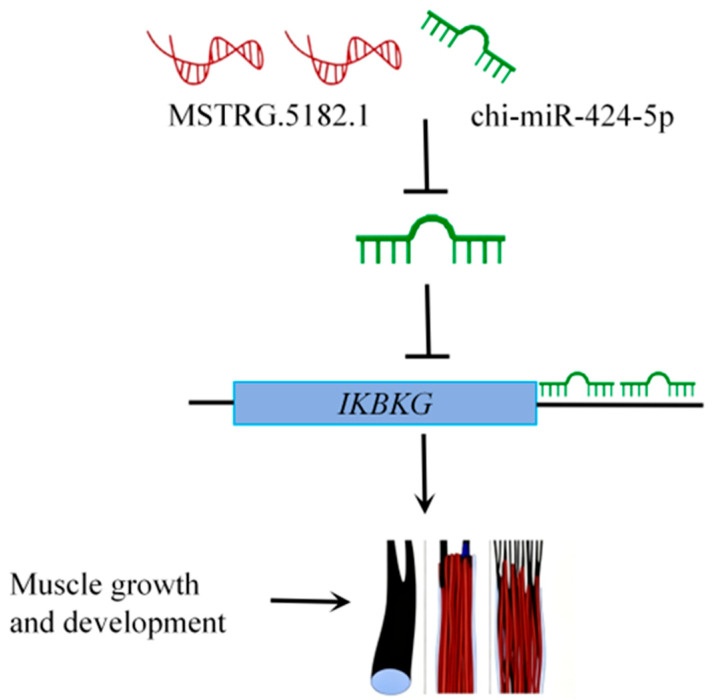
LncRNA–miRNA–mRNA regulatory mechanism diagram.

**Table 1 animals-15-03568-t001:** LncRNA and mRNA primer information.

Name	NCBI Login Number	Primer Sequence (5′ → 3′)	Fragment Size, bp
β-Actin		F: GGACTTCGAGCAGGAGATGG	104
		R: CCAGGAAGGAAGGCTGGAAG	
IKBKG	XM_013976823.2	F: CTGAAGACATGCCAGCAGATG	126
		R: CAAAGCCTGGCGCTCCTTAG	
MLXIP	XM_018061302.1	F: TTTAAAGCAGAACCGGCAGAT	165
		R: TGCAGCTTGGTGATGTACTCC	
FBXO32	XM_005688865.3	F: AAGCTGATCCATGCGAGTGT	100
		R: CCTTTCCTCAAACGCTGTGC	
MSTRG.5182.1		F: GTTTACAGGGAGCAGAGCGT	157
		R: TAACCGTTCTGCAGGGACAC	
MSTRG.5992.3		F: GCTCCTGGAATTGGGGTCTC	197
		R: CAGATTGGGGGAAGCAGAGG	
MSTRG.7890.1		F: TCCCCATAGACTGAGGAGCC	223
		R: CACAGGTGGGAGAGTAGGGA	

The primers listed in this table are intended for use in PCR with cDNA synthesized from RNA.

**Table 2 animals-15-03568-t002:** MiRNA primer information.

Name	Primer Type	Primer Sequence (5′ → 3′)	Fragment Size, bp
U6	F	GGAACGATACAGAGAAGATTAGC	68
	R	TGGAACGCTTCACGAATTTGCG	
chi-miR-424-5p	SL-RT	GTCGTATCCAGTGCAGGGTCCGAGGTATTCGCACTGGATACGACTTTTGA	-
	F	GCGCCAGCAGCAATTCATG	65
	R	GTCGTATCCAGTGCAGGGT	
chi-miR-22-3p	SL-RT	GTCGTATCCAGTGCAGGGTCCGAGGTATTCGCACTGGATACGACAAGAAC	-
	F	GCGCAAGCTGCCAGTTG	67
	R	GTCGTATCCAGTGCAGGGT	
chi-miR-145-3p	SL-RT	GTCGTATCCAGTGCAGGGTCCGAGGTATTCGCACTGGATACGACGTTCTT	-
	F	GCGCGCATTCCTGGAAATACT	71
	R	GTCGTATCCAGTGCAGGGT	

“SL-RT” refers to ‘stem–loop reverse transcription primers’ used for miRNA reverse transcription into cDNA. Other primers listed are for PCR amplification from mRNA-derived cDNA.

**Table 3 animals-15-03568-t003:** Sequencing data quality assessment.

Sample	Raw Reads	Clean Reads	Error Rate	Q20 /%	Q30 /%	GC Content	Total Mapped	Mapped Ratio
CG-1	90,671,620	90,270,588	0.06	98.00	94.27	56.96	89,631,538	83.02
CG-2	95,248,262	94,803,262	0.09	97.77	93.77	57.31	94,367,874	78.39
CG-3	97,739,300	97,350,874	0.06	97.90	94.04	56.91	96,925,510	82.34
CG-4	109,992,834	109,499,310	0.07	97.78	93.81	57.91	108,985,938	78.03
CG-5	88,231,856	87,945,132	0.05	97.94	94.04	57.69	87,572,092	76.57
CG-6	96,761,602	96,367,518	0.06	97.90	94.04	57.96	95,923,486	77.52
CG-7	77,733,584	77,478,014	0.05	97.98	94.16	56.49	77,079,874	76.81
CG-8	81,516,648	81,160,334	0.06	97.73	93.67	56.32	80,736,506	76.29
CG-9	80,439,174	80,129,922	0.04	97.89	94.04	57.31	79,693,318	78.01
CG-10	90,861,758	90,559,984	0.06	98.19	94.62	58.86	90,093,198	84.88
CG-11	80,538,374	80,276,974	0.06	98.14	94.48	58.22	79,957,986	86.23
CG-12	83,032,174	82,815,918	0.05	98.30	94.83	58.50	82,474,292	82.14

Raw Reads—initial sequences from the sequencer; Clean Reads—high-quality sequences after quality control; Error Rate—proportion of incorrectly identified bases; Q20—proportion of bases with Phred quality score ≥ 20; Q30—proportion of bases with Phred quality score ≥ 30; GC Content—percentage of G and C bases in the sequence; Total Mapped—number of base pairs (bp) or reads aligned to the reference genome; and Mapped Ratio—ratio of mapped reads to total reads.

## Data Availability

The original contributions presented in this study are included in the article. Further inquiries can be directed to the corresponding author.

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
