# Peer review of "Construction and Functional Analysis of the ceRNA Regulatory Network Associated with Muscle Development in Shaanbei White Cashmere Goats"

_animals, 2025, doi:10.3390/ani15243568_

Round 1
Reviewer 1 Report
Comments and Suggestions for Authors
Long non-coding RNAs (lncRNAs) are increasingly recognized as key regulators in a wide range of molecular and cellular processes. In this study, the authors performed RNA-seq on skeletal muscle from cashmere goats at different developmental stages, with a focus on characterizing lncRNA expression. They report the identification of a novel lncRNA, MSTRG.5182.1, which adds valuable insight into muscle development in this species. These findings are of great significance not only for agricultural breeding and production, but also for fundamental biological research.
Comments:
The current title is overstated. To conclusively claim that “MSTRG.5182.1 regulates muscle formation in goats through the chi-miR-424-5p/IKBKG pathway,” additional biological evidence is required—such as MSTRG.5182.1 knockout or loss-of-function experiments, as well as epistasis analyses involving MSTRG.5182.1 and the chi-miR-424-5p/IKBKG axis. At present, the study only provides gene-expression–based correlations. Therefore, the title should be toned down to reflect the data actually presented.
In Table 1, the title “LncRNA and mRNA Primer Information” may be misleading, as it implies that the primers were used to amplify RNA directly. In practice, these primers should be used for PCR with cDNA synthesized from RNA. Please clarify this in the table title or legend to avoid confusion.
Table 2. What is “SL-RT”? Expected fragment size for U6 is missing.
Table 3 requires additional clarification of several items. For example, what are Clean reads (from raw reads), Q20, Q30, Total mapped (bp or reads)?
Figure 1 needs clearer explanations of several abbreviations. For instance, what are “PCA” “CPC2, CNCI, and FEELnc”? In the legend, why does every word begin with an all-uppercase initial?
Figure 2.What does “Cell” in D mean? The items listed beneath it represent diverse, cell-related categories.
Figure 3 focuses on miRNA and ceRNA interactions, but it does not include mRNA-level analysis. Since lncRNAs can also directly regulate gene expression at the mRNA level, it would be helpful to either include mRNA-related data or clarify in the text why mRNA was not analyzed in this context.In Panel C, please replace the Chinese title with English.
Figure 4 and Figure 5 could be combined into a single figure since they address similar aspects of the same analysis.
Figure 6. including a simple schematic illustrating the regulatory relationships between the lncRNA, miRNA, and mRNA would help readers understand the results and the proposed mechanism.
Figure 7. For panels E and F, it would be important to show the mRNA levels of the luciferase gene. This would help distinguish whether the observed changes in luciferase activity are due to transcriptional regulation or other post-transcriptional effects.
Reviewer 2 Report
Comments and Suggestions for Authors
Long non coding RNAs (lnRNAs) play an important role in regulating the growth traits of agricultural animals. This study aims to explore the lnRNAs related to meat production traits in cashmere goats. The author analyzed samples of the longest dorsal muscle of cashmere goats at four growth stages (gestational age of 120 days, 1 month, 3 months, and 10 months) and identified a total of 3480 lncRNAs, of which 1141 were newly discovered. In depth analysis revealed that MSTRG.5182.1 plays a critical role in the proliferation and differentiation of muscle cells, and further research has shown that MSTRG.5182.1 affects muscle growth by regulating the chi-miR-424-5p/KBKG signaling pathway.
This study provides potential molecular targets for improving the quality and production performance of lamb meat, and provides a data basis for further molecular breeding and meat quality improvement of cashmere goats.
Overall, the experimental design of this study is reasonable, the analysis is thorough and appropriate, and innovative results have been achieved, with the potential for professional transformation.
In view of this, the reviewer suggests that the editor may consider accepting the publication of the class paper. At the same time, there are some minor shortcomings in the manuscript, and it is recommended that the author refer to them for revision:
(1) The research sample did not fully disclose information such as gender, variety, health status, and feeding conditions, which are crucial for the growth status of agricultural animals and even directly affect the expression of long non coding RNAs. Suggest adding a table to display the sample situation in detail;
(2) There are three types of non English text in Figure 3 that need to be corrected;
(3) Figure 5A: It is recommended to analyze and approve the data statistics again. Based on the reviewer's experience, there are doubts about the differences between individual data;
(4) The text font in Figure 6C is too large and does not match the ones in A and B. It is recommended to make corrections.
Reviewer 3 Report
Comments and Suggestions for Authors
Dear authors,
I will recommend your article for publication, but I have several questions about the research results. They are short, but very serious to be asked.
L69 – Need reference
L107 – All goats are male?
L196 – Last column is primer size, not fragment?
L219 – What is transfection reagent?
L234 – In this part all using methods must be declared. An example, PCA.
L252 – This table is not necessary. You can add average values in text.
L316 – Some symbols in figure too small. I recommend to separate on two.
L354 – Part C – correct legend
L447 – Part A is too small. I recommend enlarge it to single figure.
L495 – Add reference
Regards,
Reviewer 4 Report
Comments and Suggestions for Authors
-
The study is well-designed and contributes meaningful insights into the regulatory roles of lncRNAs in goat muscle development. Only a few small issues need attention before acceptance.
-
The manuscript would benefit from a careful line-by-line proofreading to correct minor grammatical slips, inconsistent spacing, and a few long sentences that reduce readability.
-
The Simple Summary is clear but slightly wordy; shortening a couple of sentences would improve flow.
-
There is an inconsistency in the naming of the key lncRNA (MSTRG.5182.1 vs. MSTRG.5182.12). Please verify and use a single, correct name throughout.
-
In the Methods section, briefly mention the RNA integrity threshold (e.g., RIN value) used before sequencing to strengthen transparency.
-
The transfection procedure is very detailed; you may condense it slightly so it reads less like a step-by-step protocol and more like a methods description.
-
Figures 1–3 contain useful information, but the legends should clarify what each sample code (e.g., CG-A-1, CG-B-3) represents so readers do not have to refer back to earlier sections.
-
The color contrast of Figure 3B could be improved for better visibility of upregulated versus downregulated miRNAs.
-
In Figure 7, clearly labeling WT and MUT groups within the panels will avoid confusion.
-
The Discussion is strong, but a few ideas are repeated in different paragraphs; removing redundancies will tighten the narrative.
-
Including a brief comment on limitations (e.g., absence of in vivo functional verification) would create a more balanced discussion.
-
A short comparison with findings from similar lncRNA studies in other goat breeds would help position the work within the broader literature.
-
Ensure that reference formatting is consistent—some journal names, spacing, and DOI formats need minor correction.
-
Overall, this is a solid manuscript requiring only small textual and formatting improvements. After addressing these points, it should be ready for publication.
